# Continuous versus Intermittent Enteral Tube Feeding for Critically Ill Patients: A Prospective, Randomized Controlled Trial

**DOI:** 10.3390/nu14030664

**Published:** 2022-02-04

**Authors:** Hong-Yeul Lee, Jung-Kyu Lee, Hye-Jin Kim, Dal-Lae Ju, Sang-Min Lee, Jinwoo Lee

**Affiliations:** 1Department of Critical Care Medicine, Seoul National University Hospital, Seoul 03080, Korea; takumama@naver.com (H.-Y.L.); sangmin2@snu.ac.kr (S.-M.L.); 2Division of Respiratory and Critical Care, Department of Internal Medicine, Seoul Metropolitan Government—Seoul National University Boramae Medical Center, 20, Seoul 07061, Korea; jk1909@empas.com; 3Department of Food Service and Nutritional Care, Seoul National University Hospital, Seoul 03080, Korea; isof925@snuh.org; 4Department of Nutrition, Seoul Metropolitan Government—Seoul National University Boramae Medical Center, Seoul 07061, Korea; jurea@snuh.org; 5Division of Pulmonary and Critical Care Medicine, Department of Internal Medicine, Seoul National University College of Medicine, Seoul 03080, Korea

**Keywords:** critically ill, enteral feeding, mechanical ventilation, nutrition

## Abstract

The appropriate strategy for enteral feeding remains a matter of debate. We hypothesized that continuous enteral feeding would result in higher rates of achieving target nutrition during the first 7 days compared with intermittent enteral feeding. We conducted an unblinded, single-center, parallel-group, randomized controlled trial involving adult patients admitted to the medical intensive care unit who required mechanical ventilation to determine the efficacy and safety of continuous enteral feeding for critically ill patients compared with intermittent enteral feeding. The primary endpoint was the achievement of ≥80% of the target nutrition requirement during the first 7 days after starting enteral feeding. A total of 99 patients were included in the modified intention-to-treat analysis (intermittent enteral feeding group, *n* = 49; continuous enteral feeding group, *n* = 50). The intermittent enteral feeding group and continuous enteral feeding group received 227 days and 226 days of enteral feeding, respectively. The achievement of ≥80% of the target nutrition requirement occurred significantly more frequently in the continuous enteral feeding group than in the intermittent enteral feeding group (65.0% versus 52.4%, respectively; relative risk, 1.24; 95% confidence interval, 1.06–1.45; *p* = 0.008). For patients undergoing mechanical ventilation, continuous enteral feeding significantly improved the achievement of target nutrition requirements.

## 1. Introduction

Major clinical practice guidelines emphasize the importance of early enteral nutrition (EN) for critically ill patients. Unfortunately, many critically ill patients are underfed, and only approximately 40–60% of the recommended nutritional goals are met [1,2]. Undernutrition has been associated with an increased risk of complications such as nosocomial infections and mortality [3,4,5]. These deficits were highest during the first week of admission and were not fully balanced during the remaining intensive care unit (ICU) stay [6]. Inevitably, large gaps exist between guidelines and real-world practices [7]; therefore, more evidence is needed to establish the optimal method of delivering EN and to close those gaps.

EN can be administered using various methods. An hourly rate of EN was administered using a feeding pump during continuous feeding. However, EN was administered for 20–60 min every 4–6 h during intermittent feeding. Continuous feeding may be associated with reductions in diarrhea [8] and aspiration [9]; however, feeding pumps are always required, and feeding is frequently interrupted for those needing medications or procedures. Intermittent feeding is theoretically more physiologic, does not require a feeding pump, and may increase protein synthesis; however, it may increase the workload of ICU nurses [10]. Moreover, the selection of strategies of enteral feeding may be influenced by several clinical and organizational factors, such as the type of EN formulas, viscosity of EN formulas, glucose variability, gastrointestinal intolerance, chrononutrition, refeeding syndrome, and scheduled visits to the imaging department or operating room [11,12,13]. Despite the lack of supporting evidence to make strong recommendations for its use, continuous feeding has been favored over intermittent feeding [8,14,15].

This study aimed to test the hypothesis that continuous enteral feeding would result in a higher rate of achieving target nutrition during the first 7 days compared with intermittent enteral feeding.

## 2. Materials and Methods

### 2.1. Study Design, Setting, and Ethical Considerations

We conducted an unblinded, single-center, parallel-group, randomized controlled trial. The main objective of this study was to determine the efficacy and safety of continuous enteral feeding for critically ill patients compared with intermittent enteral feeding. Patients in a medical ICU at Seoul National University Hospital, which is a 1778-bed tertiary-care referral hospital in South Korea, were enrolled from May 2014 to December 2019; they were followed-up for 28 days or until ICU discharge. The Institutional Review Board of Seoul National University Hospital approved the study and protocol (approval number IRB-H-1403-124-568). All patients or their legal representatives provided written informed consent to participate in the study. This trial was registered with ClinicalTrials.gov (number NCT02159456).

### 2.2. Participants

All consecutive adult patients admitted to the medical ICU who required mechanical ventilation underwent screening before enrollment. The inclusion criteria were assessed within 48 h of the ICU admission. Patients who were at least 20 years of age were eligible for inclusion if they were receiving mechanical ventilation, were expected to require ventilation for ≥48 h, and were to start enteral feeding within 48 h of the ICU admission. Patients were excluded if they had a history of hypersensitivity reactions to prokinetics, had gastrointestinal bleeding, had bowel obstruction, had refractory vomiting or diarrhea, had a history of seizure or pheochromocytoma, had enterostomy or gastrostomy in situ, experienced difficulty with insertion or maintenance of a nasogastric tube, needed a specialized feeding regimen (such as diets for hemodialysis, chronic renal failure, or diabetes), had undergone abdominal surgery within 1 month, or were pregnant.

### 2.3. Randomization

Patients who were enrolled by the ICU attending physician or fellow were randomized in a 1:1 ratio to receive either continuous or intermittent enteral feeding. We used a permuted block randomization scheme with randomly selected block sizes ranging from three to six created by the Medical Research Collaborating Center (stratified according to age and presence of shock). Throughout the study, an independent research nurse maintained the randomization list using sequentially numbered, opaque, sealed envelopes that were inaccessible to clinical investigators. As the method of administering enteral feeding cannot be masked, blinding of the healthcare providers was not feasible.

### 2.4. Protocol

In both groups, enteral feeding commenced within 48 h of the ICU admission. The position of the nasogastric tube was confirmed by chest radiography before beginning enteral feeding. The target nutrition goal was calculated by a dedicated nutritionist who was blinded to the groups. The volume of enteral feeding was gradually increased to reach 100% of the target nutrition requirement at the same time point, regardless of the allocated group. In both groups, the enteral feeding algorithm was designed to reach 100% of the target nutrition requirement within 48 h after commencing enteral feeding (Appendix A). All patients received the same enteral feeding formula. Patients were treated with chlorhexidine mouthwash three times daily and placed in the head-up position (≥30°). As many of the enrolled patients were critically ill patients with various comorbidities, various medications were administered via both nasogastric tube and intravenous route.

Patients in the intermittent enteral feeding group received enteral feeding through a nasogastric tube at 9:00, 13:00, 17:00, and 21:00. Each feeding was completed within 1 h using a gravity-based infusion. The gastric residual volume was measured with a 50 mL syringe before each feeding. A gastric residual volume less than 250 mL was returned to the stomach, and the excess volume was discarded. The initial enteral feeding volume was 150 mL; however, the volume was adjusted according to the gastric residual volume and gastrointestinal intolerance. If two consecutive measurements indicated that the patients had a gastric residual volume less than 250 mL, then the enteral feeding volume was advanced to the next step until the target nutrition requirement was achieved. If the patients had a gastric residual volume ≥ 250 mL, then the enteral feeding volume remained unchanged until the next feeding and metoclopramide (10 mg every 8 h) was administered for 48 h. If two consecutive measurements indicated that the patients had a gastric residual volume ≥ 250 mL, then the enteral feeding volume was returned to the previous step. If the patients vomited, then enteral feeding was interrupted until the next scheduled feeding time and metoclopramide (10 mg every 8 h) was administered for 48 h. Then, enteral feeding was resumed with a volume of 150 mL. The detailed enteral feeding method is presented in Appendix A.

Patients in the continuous enteral feeding group received enteral feeding through a nasogastric tube with an infusion time of 24 h per day. The gastric residual volume was measured every 6 h with a 50 mL syringe. A gastric residual volume less than 250 mL was returned to the stomach, and the excess volume was discarded. The initial rate of enteral feeding was 25 mL/h; this rate was adjusted according to the gastric residual volume and gastrointestinal intolerance. If the patients had a gastric residual volume less than 250 mL, then the rate of infusion was increased by 25 mL/h until the target nutrition rate was achieved. If the patients had a gastric residual volume ≥ 250 mL, then the rate of infusion remained unchanged until the next measurement of the gastric residual volume and metoclopramide (10 mg every 8 h) was administered for 48 h. If two consecutive measurements indicated that the gastric residual volume was ≥250 mL, then the infusion rate was decreased by 25 mL/h. If three consecutive measurements indicated that the patients had a gastric residual volume ≥ 250 mL, then enteral feeding was interrupted until the next measurement of the gastric residual volume. If the patients vomited, then enteral feeding was interrupted until the next measurement of the gastric residual volume and administration of metoclopramide (10 mg every 8 h) was considered for 48 h. Then, enteral feeding was resumed at a rate of 25 mL/h. The detailed enteral feeding method is presented in Appendix A.

Trial enteral feeding was administered for up to 7 days or until the patient was discharged from the ICU, discontinued enteral feeding, or died (whichever occurred first). Trial enteral feeding was stopped if the patient started oral nutrition, if the patient received parenteral nutrition, if the patient met the predefined criteria (which was the same as the exclusion criteria), or if the physician decided it was in the best interest of the patient to discontinue enteral feeding. During the study period, all patients were screened and monitored for adverse events including refeeding syndrome by daily laboratory and clinical assessment.

### 2.5. End Points

The primary endpoint was the rate at which ≥80% of the target nutrition requirement was achieved during the first 7 days after the start of enteral feeding. The achievement rate was calculated as the number of days in which ≥80% of the target nutrition requirement was achieved divided by the total number of days of feeding. Secondary endpoints included ICU mortality, 28-day mortality, length of stay (LOS) in the ICU and hospital, gastrointestinal intolerance during the first 7 days after the start of enteral feeding (diarrhea, constipation, vomiting or regurgitation, abdominal pain or distension, aspiration, or receiving prokinetic drugs), days free from mechanical ventilation, days free from dialysis, and days free from vasopressor support between the time of randomization and day 28. Diarrhea, constipation, vomiting, and aspiration were defined as more than 200 g of stool/day, more than 3 days without stool, gastric content detected in the oropharynx or outside the mouth, and evidence of food material in the airway, respectively.

### 2.6. Statistical Analysis

Before initiation of the present study, a valid estimation of our primary outcome variable in critically ill patients was lacking. However, two previous studies reported that the percentage of patients who received enteral nutrition at least 80% of their estimated calorie requirements in the critical care setting was 50 to 60% [2,16]. Based upon previous studies, we assumed that an estimated baseline rate of achieving ≥80% of the target nutrition requirement during the first 7 days after the start of enteral feeding would be 56% and that patients would stay in the ICU for more than 5 days. Thus, we estimated that a sample of 102 patients (512 days of enteral feedings) would provide the study with at least 80% power and a 5% type I error (two-sided tests) to recognize an improvement of at least 20% in rate of achieving ≥80% of the target nutrition requirement in the continuous enteral feeding group. This would result in an increase from 148 to 178 achievement days per total days of feeding. We assumed a dropout rate of 10% and aimed to include 112 patients.

All analyses were conducted according to the modified intention-to-treat principle and included all randomized patients who received allocated enteral feeding. Categorical variables are reported as frequencies and percentages. Continuous variables are expressed as medians and interquartile ranges or as means and standard deviations. Regarding the baseline characteristics and clinical outcomes, differences between groups were assessed using the chi-square test or Fisher’s exact test for categorical variables and the Mann–Whitney U test or Student’s *t*-test for continuous variables. There were no missing values for the primary and secondary outcomes. All analyses were conducted using a two-sided alpha level of 0.05. All analyses were performed using IBM SPSS Statistics (version 25.0 for Windows; IBM Corp., Armonk, NY, USA).

## 3. Results

### 3.1. Participants of the Study

From May 2014 to December 2019, a total of 1078 patients were assessed for eligibility. A total of 112 patients were enrolled and randomly assigned to either intermittent enteral feeding (56 patients) or continuous enteral feeding (56 patients). Thirteen patients were subsequently excluded; therefore, 99 patients (49 in the intermittent enteral feeding group and 50 in the continuous enteral feeding group) were included in the modified intention-to-treat analysis (Figure 1).

### 3.2. Participant Characteristics

Baseline demographic and clinical characteristics of the patients are shown in Table 1. The mean age was 66.9 ± 11.5 years, 66.7% of the patients were men, the mean body mass index was 22.6 ± 3.9 kg/m^2^, 36.4% had cardiovascular disease, 30.3% had diabetes mellitus, 44.4% were diagnosed with malignancy, the mean Sequential Organ Failure Assessment (SOFA) score was 8.8 ± 4.3, and the mean Acute Physiologic Assessment and Chronic Health Evaluation II (APACHE II) score was 28.2 ± 8.6. The primary reason for mechanical ventilation was respiratory failure (82.8%). The median time from ICU admission to randomization was 15.9 h (interquartile range (IQR), 7.2–25.7 h) for the intermittent enteral feeding group and 15.9 h (IQR, 7.5–23.4 h) for the continuous enteral feeding group (*p* = 0.552). There was no difference in the calculated energy targets of the intermittent enteral feeding group and continuous enteral feeding group (24.1 ± 2.9 kcal/kg/day (1380 ± 172 kcal/day) versus 24.7 ± 2.9 kcal/kg/day (1426 ± 201 kcal/day), respectively; *p* = 0.309). At the time of randomization, vasopressors were used for 49.5% of the patients, renal replacement therapy was used for 17.2%, and systemic corticosteroid therapy was used for 62.6%. These percentages did not differ significantly between groups.

### 3.3. Primary Outcome

Enteral feeding was performed for a total of 227 days for the intermittent enteral feeding group and for a total of 226 days for the continuous enteral feeding group. The number of days during which ≥80% of the target nutrition requirement was achieved was significantly higher for the continuous enteral feeding group than for the intermittent enteral feeding group (65.0% (147 of 226 days) versus 52.4% (119 of 227 days), respectively; relative risk, 1.24; 95% confidence interval, 1.06–1.45; *p* = 0.008) (Table 2). The percentage of delivered target nutrition was significantly higher for the continuous enteral feeding group than for the intermittent enteral feeding group (91.0% (IQR, 68.1–100%) versus 83.9% (IQR, 55.6–100%), respectively; *p* = 0.033) (Figure 2). There was no difference in the median duration of target nutrition achievement between the intermittent enteral feeding group and continuous enteral feeding group (2 days (IQR, 0–5 days) versus 2 days (IQR, 1–6 days), respectively; *p* = 0.123).

### 3.4. Secondary Outcomes

The ICU mortality rates were 49.0% (24 of 49 patients) for the intermittent enteral feeding group and 32.0% (16 of 50 patients) for the continuous enteral feeding group (relative risk, 0.65; 95% confidence interval, 0.40–1.07; *p* = 0.129). Similarly, there were no significant differences between groups in terms of 28-day mortality. The hospital LOS and ICU LOS did not differ significantly between groups. There were no significant differences between the intermittent enteral feeding and continuous enteral feeding groups in terms of diarrhea (44.9% versus 44.0%; *p* > 0.999), constipation (44.9% vs. 56.0%; *p* = 0.366), vomiting or regurgitation (16.3% versus 14.0%; *p* = 0.966), abdominal pain or discomfort (16.3% versus 6.0%; *p* = 0.189), use of prokinetic drugs (24.5% versus 38.0%; *p* = 0.218), and aspiration (0.0% versus 4.0%; *p* = 0.484). The number of days free from mechanical ventilation, dialysis, and vasopressor support did not differ significantly between groups. No significant harm or unintended effects including refeeding syndrome were observed in either group. We identified 28 patients (28.3%) with hypophosphatemia (drop below 2 mg/dL within 72 h after the start of enteral feeding); however, there was no difference in hypophosphatemia incidence between the intermittent enteral feeding group and continuous enteral feeding group (15 patients (30.6%) versus 13 patients (26.0%), respectively; *p* = 0.774). There were no events of severe hypophosphatemia (drop below 1 mg/dL within 72 h after the start of enteral feeding). None of the patients discontinued or changed enteral feeding due to hypophosphatemia.

## 4. Discussion

During this randomized trial, we compared continuous enteral feeding with intermittent enteral feeding for adult patients who required mechanical ventilation and were admitted to the medical ICU. The strategy of continuous enteral feeding significantly improved the achievement of ≥80% of the target nutrition requirement compared with intermittent enteral feeding. However, there were no differences between intermittent enteral feeding and continuous enteral feeding in terms of mortality or other key secondary outcomes, including hospital and ICU LOS, gastrointestinal intolerance, and organ support.

Previous randomized clinical trials have shown inconsistent results regarding which strategy is the most appropriate for reaching the target nutrition requirement and involves fewer complications [17,18,19,20,21]. Two previous randomized clinical trials reported that intermittent or bolus enteral feeding required more time to reach the target nutrition requirement and appeared to provoke more gastrointestinal intolerance, such as high gastric residual volumes, vomiting, and diarrhea, than continuous enteral feeding [17,18]. During a previous randomized clinical trial including 50 patients with percutaneous endoscopic gastrostomy, there were no differences between bolus enteral feeding and continuous enteral feeding in terms of glycemic variability, insulin utilization, incidence of hypoglycemia, or time to the nutritional delivery goal of ≥80% [19]. One recent randomized clinical trial reported that intermittent enteral feeding results in greater achievement of the target nutrition requirement than continuous enteral feeding [21]. During the present study, the rate of achieving ≥80% of the target nutrition requirement was significantly higher for those administered continuous enteral feeding than for those administered intermittent enteral feeding. Several factors may explain why the results are inconsistent across studies, including heterogeneity of patients selected for inclusion and interventions, choice of outcome endpoints and trial design, and differences in sample size. One strength of our study was that the volume of enteral feeding was gradually increased to reach the target nutrition requirement at the same time point regardless of the allocated group. Another strength was that all patients received the same enteral feeding formula. Therefore, our strict protocolized administration of enteral feeding made it possible to accurately compare continuous and intermittent enteral feeding in terms of the primary outcome. Further large, well-designed studies are warranted to confirm these inconsistent results.

The current guidelines recommend early enteral feeding within 48 h for critically ill adult patients who require nutritional support therapy [8,14,15]. Additionally, although these guidelines recommend continuous enteral feeding rather than intermittent enteral feeding, the available evidence is limited by heterogeneity and clinical variability across studies. One meta-analysis showed that continuous enteral feeding was associated with a reduction in diarrhea but that no association was identified for other outcomes, including mortality, morbidity, and glycemic variability [8]; however, this analysis only included five randomized clinical trials. Another more recent meta-analysis showed no differences in diarrhea [22]. During the present study, there were no significant differences between the intermittent enteral feeding group and continuous enteral feeding group in terms of diarrhea (44.9% versus 44.0%; *p* > 0.999). Moreover, the incidence of gastrointestinal intolerance, including constipation, vomiting or regurgitation, abdominal pain or discomfort, use of prokinetic drugs, and aspiration, did not differ significantly between groups. Additionally, there were no differences in mortality, ICU LOS, hospital LOS, and the number of days free from organ support between groups. The strategy of intermittent enteral feeding, theoretically, may provide physiological and metabolic benefits over continuous enteral feeding, such as the improvement of protein synthesis, preservation of the circadian rhythm, maintenance of the entero-hormonal response to luminal nutrients, and activation of autophagy [10,11,23]. During the acute phase of critical illness, the freedom to actively participate in rehabilitation therapy; no delays in procedures, tests, and medications that requires empty stomach; and the relatively inexpensive cost were also important advantages of intermittent feeding [11]. During a recent phase 2 clinical trial including 121 mechanically ventilated adult patients with multiorgan failure, intermittent enteral feeding during early critical illness did not preserve muscle mass compared with continuous enteral feeding [21]. Adequately powered randomized clinical trials are needed to explore these theoretical benefits. The results of this study support the current guidelines recommending continuous enteral feeding as the optimal strategy for EN [8,14,15].

Our study had several limitations. First, we could not blind the ICU physicians or participants to the treatment allocation. The nature of the treatments precluded the masking of the physicians and participants. Second, we only included patients admitted to the medical ICU. This may have limited the generalizability of our results to trauma or surgical patients. Third, the target nutrition requirements of this study were assessed by a dedicated nutritionist who was blinded to the groups. Although indirect calorimetry is the gold standard for determining the energy requirements of critically ill patients [8,14], evidence regarding the widespread implementation of indirect calorimetry in the ICU is lacking [24]. Fourth, although we excluded patients who needed specialized feeding regimen or with conditions inappropriate for study participation, additional clinical and organizational factors affecting the selection of enteral feeding strategies may have been overlooked [11,12,13]. However, we did not allow crossover between the groups during the study period.

## 5. Conclusions

In conclusion, the strategy of continuous enteral feeding significantly improved the achievement of target nutrition requirements compared with the strategy of intermittent enteral feeding. However, there were no differences between intermittent enteral feeding and continuous enteral feeding in terms of mortality or other key secondary outcomes, including hospital and intensive care unit length of stay, gastrointestinal intolerance, and organ support. The results of this study support the current guidelines recommending continuous enteral feeding as the optimal strategy for EN.

## Figures and Tables

**Figure 1 nutrients-14-00664-f001:**
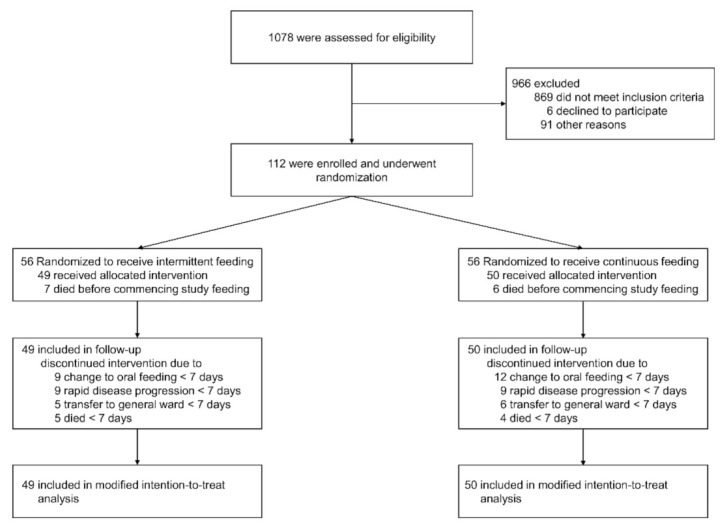
Patient recruitment flow diagram.

**Figure 2 nutrients-14-00664-f002:**
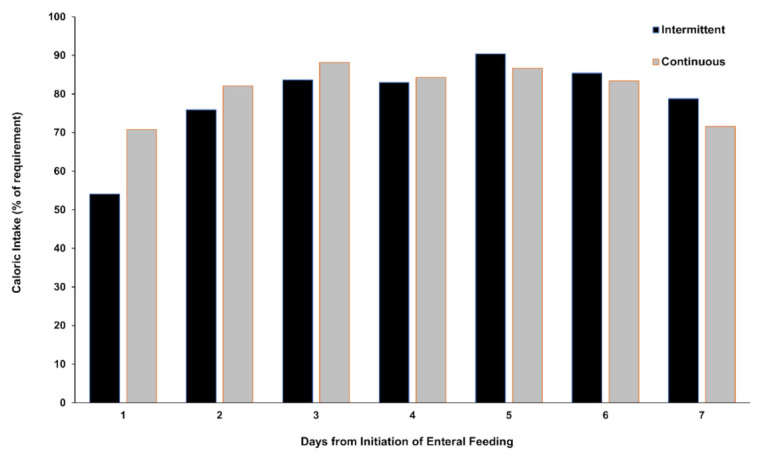
Daily enteral nutrition delivery during the 7-day trial period. This figure shows the percentage of delivered target enteral nutrition.

**Table 1 nutrients-14-00664-t001:** Baseline characteristics of the patients according to study group.

	Intermittent Feeding (*n* = 49)	Continuous Feeding (*n* = 50)	*p*-Value
Age, years	66.2 ± 12.7	67.5 ± 10.3	0.565
Sex, male, *n* (%)	33 (67.3)	33 (66.0)	>0.999
Body mass index, kg/m^2^	22.0 ± 3.9	23.3 ± 3.9	0.107
Primary diagnosis, *n* (%)			0.210
Respiratory failure	39 (79.6)	43 (86.0)	
Cardiac arrest	6 (12.2)	2 (4.0)	
Sepsis or septic shock	2 (4.1)	4 (8.0)	
Other	2 (4.1)	1 (2.0)	
Comorbidities, *n* (%)			
Cardiovascular disease	16 (32.7)	20 (40.0)	0.582
Diabetes mellitus	15 (30.6)	15 (30.0)	>0.999
Chronic lung disease	21 (42.9)	17 (34.0)	0.484
Chronic kidney disease	7 (14.3)	10 (20.0)	0.626
Chronic liver disease	7 (14.3)	5 (10.0)	0.730
Malignancy	21 (42.9)	23 (46.0)	0.911
Immunodeficiency	21 (42.9)	19 (38.0)	0.774
Chronic neurologic disease	6 (12.2)	5 (10.0)	0.972
Median time from ICU admission to randomization (IQR), h	15.9 (7.2–25.7)	15.9 (7.5–23.4)	0.552
APACHE II score	27.7 ± 9.3	28.6 ± 8.0	0.620
SOFA score	9.1 ± 4.4	8.6 ± 4.1	0.560
Ongoing treatments at randomization, *n* (%)			
Mechanical ventilation	49 (100)	50 (100)	>0.999
Renal replacement therapy	9 (18.4)	8 (16.0)	0.963
Vasopressor therapy	26 (53.1)	23 (46.0)	0.616
Systemic corticosteroid therapy	34 (69.4)	28 (56.0)	0.242
Anti-infectious treatment	47 (95.9)	50 (100)	0.466
Sedative drugs	29 (59.2)	31 (62.0)	0.935
Analgesic drugs	45 (91.8)	45 (90.0)	>0.999
Energy target (kcal/day)	1380 ± 172	1426 ± 201	0.220
Energy target per ideal body weight (kcal/kg/day)	24.1 ± 2.9	24.7 ± 2.9	0.309
Laboratory variables			
Serum creatinine, mg/dL	1.06 (0.64–1.85)	1.08 (0.78–1.76)	0.788
Lactate, mmol/L	2.2 (1.6–3.4)	2.0 (1.5–3.9)	0.841
C-reactive protein, mg/dL	13.4 (7.6–22.5)	13.3 (5.0–25.7)	0.975
Serum albumin, mg/dL	2.8 (2.4–3.0)	2.6 (2.3–3.0)	0.586
Glucose, mg/dL	167 (131–220)	172 (115–237)	0.804

APACHE = Acute Physiologic Assessment and Chronic Health Evaluation; ICU = intensive care unit; IQR = interquartile range; SOFA = Sequential Organ Failure Assessment.

**Table 2 nutrients-14-00664-t002:** Primary and secondary outcomes according to the modified intention-to-treat analysis.

	Intermittent Feeding (*n* = 49)	Continuous Feeding (*n* = 50)	Relative Risk (95% CI)	*p*-Value
Primary outcome				
≥80% of the target requirement, achievement days/total days of feeding (%)	119/227 (52.4)	147/226 (65.0)	1.24 (1.06–1.45)	0.008
Secondary outcomes				
ICU mortality, *n* (%)	24 (49.0)	16 (32.0)	0.65 (0.40–1.07)	0.129
Death within 28 days, *n* (%)	26 (53.1)	21 (42.0)	0.79 (0.52–1.20)	0.368
Median length of hospital stay (IQR), days				
In-hospital survivors	25 (17–33)	22 (11–38)		0.763
In-hospital nonsurvivors	9 (4–15)	10 (6–19)		0.603
Median length of ICU stay (IQR), days				
ICU survivors	8 (5–11)	6 (3–11)		0.443
ICU nonsurvivors	6 (3–10)	7 (4–12)		0.782
Gastrointestinal intolerance, *n* (%)				
Diarrhea	22 (44.9)	22 (44.0)	0.98 (0.63–1.52)	>0.999
Constipation	22 (44.9)	28 (56.0)	1.25 (0.84–1.85)	0.366
Vomiting or regurgitation	8 (16.3)	7 (14.0)	0.86 (0.34–2.18)	0.966
Abdominal pain or distension	8 (16.3)	3 (6.0)	0.37 (0.10–1.30)	0.189
Aspiration	0 (0)	2 (4.0)	4.90 (0.24–99.57)	0.484
Received prokinetic drugs	12 (24.5)	19 (38.0)	1.55 (0.85–2.84)	0.218
Days without mechanical ventilation ^a^	0 (0–20)	11 (0–20)		0.142
Days without dialysis ^a^	11 (0–28)	24 (7–28)		0.077
Days without vasopressor support ^a^	11 (0–24)	19 (4–26)		0.062

^a^ The number of days alive and free from mechanical ventilation, dialysis, and vasopressor support were calculated for the first 28 study days. CI = confidence interval; ICU = intensive care unit; IQR = interquartile range.

## Data Availability

The datasets generated during the current study are available from the corresponding author upon reasonable request.

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
