# Peer review of "Continuous versus Intermittent Enteral Tube Feeding for Critically Ill Patients: A Prospective, Randomized Controlled Trial"

_nutrients, 2022, doi:10.3390/nu14030664_

Round 1

Reviewer 1 Report

Thank you for submitting the manuscript "Continuous Versus Intermittent Enteral Tube Feeding for Critically Ill Patients: A Prospective, Randomized Controlled Trial" to Nutrients. The experiment was well designed and thus the results found are valid. The subject is extremely relevant to the clinical area of ​​nutrition since few studies are carried out in this area and there really is a self-index of malnutrition in patients with tube use. I have no concerns about what was exposed because the text is clear and objective. However, before accepting the manuscript for Nutrients, some questions need to be answered and the text in a general needs to be improved.

Introduction: there are indications for the use of continuous or intermittent feeding, such as the type of diet (pasty or liquid), need for caloric intake (in cases of problems with refeeding), etc. This information needs to be added in the introduction so it doesn't look like the indication is the same.

Material and methods: this item, although well described in terms of the study protocol, needs to better add practical issues regarding the continuous and intermittent feeding system. Were all diets started at the same flow rate even in different clinical cases? This is not common practice... Even if a careful nutritionist has planned the nutritional requirement of each patient, it is necessary to provide information on the objective of diet administration, for example, how long the objective was to meet part or all of the value Calorie calculated for the patient? Was this different in each patient? Although the study was randomized, was there attention to the issue of refeeding, which is a major obstacle to obtaining enteral nutrition in patients with tube use? Another pertinent question that must be answered in this item is in relation to the medication question: were the patients undergoing treatment using medication via a tube or intravenously? Please add this information.

It seems to me that another weak point of the work is that the two systems are not necessarily indicated for the same clinical cases. I believe that a discussion of this should be incorporated into the text.

Author Response

We would like to express our sincere gratitude for the opportunity to revise our manuscript, nutrients-1582667, entitled: “Continuous versus intermittent enteral tube feeding for critically ill patients: A prospective, randomized controlled trial”. We appreciate the constructive criticisms of the reviewer. All the comments and suggestions have led to a significant improvement of our manuscript, and we have accordingly revised the text to the best of our ability.

We have highlighted all changes in the revised manuscript with yellow highlight and have addressed all referees’ comments in the point-by-point responses in the attached files. We hope that our revised manuscript is now satisfactory for full publication in “Nutrients”.

Reviewer 2 Report

Thank you for studying this important area of everyday practice. This report shows a study with a reasonable sample size and appropriate design. Some considerations need more detail and the whole article would benefit from a native English speaker proof-reading and correcting some sentence structures.

Firstly, the statistical analysis section needs some more detail. What was the basis for the hypothesis that you describe here? Was the power calculation based on this hypothesised difference of 12%? Need to make this clear in your wording. Also, note change in percent is a problematic measure that can invalidate some statistical tests and it would be better in the analysis to use raw numbers rather than a composite like percentage.

In your introduction and discussion, you have not mentioned a major difference with intermittent regimens, the amount of extra work for the nursing staff when feed has to be set up four times instead of just leaving the pump to run the feed. This component of feasibility is a key reason why some ICUs might prefer a particular regimen for feeding. Some ICUs are set up in a way that makes intermittent regimens work better, but in other ICUs it will be really inconvenient.

In the discussion, you briefly mention suggested benefits of intermittent regimens, such as 'improvement of protein synthesis, preservation of the circadian rhythm, maintenance of the entero-hormonal response to luminal nutrients, and activation of autophagy'. It is important to note here that most or all of these benefits will not occur in critical illness where these factors are all overwhelmed by the acute phase response metabolism. In the ICU setting, advantages of intermittent regimens might include allowing time for physiotherapy or medications that require an empty stomach, or transfer for tests/procedures.

Author Response

(The authors gave the same response as above.)

Reviewer 3 Report

Authors perform a comparison between continuous and intermittent enteral feeding in critically ill patients. The topic is of relevance: some authors have indicated the theoretical advantages of intermittent feeding in these patients, but the issue is controversial. The controversy is well analysed in the manuscript.

The study is well planned and adequately performed. Results are clear. Study limitations are also indicated in the manuscript.

Author Response

We sincerely thank you for taking the time to read our manuscript and for your thoughtful comments.